ecology, environmental science

collapse, climate change, Rapa Nui, overpopulation, population theory

**Author for correspondence:**
M. Lima
e-mail: mlima@bio.puc.cl

# Ecology of the collapse of Rapa Nui society

M. Lima[1,2], E. M. Gayo[2,3], C. Latorre[1,4,5], C. M. Santoro[6], S. A. Estay[2,7], N. Cañellas-Boltà[8], O. Margalef[9,12], S. Giralt[8], A. Sáez[10], S. Pla-Rabes[11,12] and N. Chr. Stenseth[13]

[1]Departamento de Ecología, Pontificia Universidad Católica de Chile, Santiago, Chile
[2]Center of Applied Ecology and Sustainability (CAPES), Pontificia Universidad Católica de Chile, Santiago, Chile
[3]Center for Climate and Resilience Research (CR2), Santiago, Chile
[4]Centro UC del Desierto de Atacama, Pontificia Universidad Católica de Chile, Santiago, Chile
[5]Institute of Ecology and Biodiversity (IEB), Santiago, Chile
[6]Instituto de Alta Investigación, Universidad de Tarapacá, Arica, Chile
[7]Instituto de Ciencias Ambientales y Evolutivas, Universidad Austral de Chile, Valdivia, Chile
[8]Institute of Earth Sciences Jaume Almera (ICTJA-CSIC), Lluís Solé Sabarís s/n, E-08028 Barcelona, Spain
[9]Consejo Superior de Investigaciones Científicas, Global Ecology Unit CREAF-CSIC-UAB, 08193 Cerdanyola del Vallès, Catalonia, Spain
[10]Department of Earth and Ocean Dynamics, Universitat de Barcelona, Marti i Franques s/n, E-08028 Barcelona, Spain
[11]BABVE, Universitat Autònoma de Barcelona (UAB), Cerdanyola del Vallès 08193, Spain
[12]Center for Ecological Research and Forestry Application (CREAF), E-08193 Cerdanyola del Vallès, Catalonia, Spain
[13]Centre for Ecological and Evolutionary Synthesis (CEES), Department of Biosciences, University of Oslo, PO Box 1066, Blindern, 0316 Oslo, Norway

ML, 0000-0002-3700-2945; SP-R, 0000-0003-3532-9466

Collapses of food producer societies are recurrent events in prehistory and have triggered a growing concern for identifying the underlying causes of convergences/divergences across cultures around the world. One of the most studied and used as a paradigmatic case is the population collapse of the Rapa Nui society. Here, we test different hypotheses about it by developing explicit population dynamic models that integrate feedbacks between climatic, demographic and ecological factors that underpinned the socio-cultural trajectory of these people. We evaluate our model outputs against a reconstruction of past population size based on archaeological radiocarbon dates from the island. The resulting estimated demographic declines of the Rapa Nui people are linked to the long-term effects of climate change on the island's carrying capacity and, in turn, on the 'per-capita food supply'.

## 1. Introduction

Prehistoric producer societies provide unique socio-ecological laboratories for understanding the relationship between human population, food production and climate variability, with profound implications for current sustainability problems [1]. Recently, a number of studies covering the Holocene period have used the summed probability densities (SPDs) of archaeological radiocarbon dates to infer demographic changes of food producer human societies and their potential connections to large-scale climatic processes [2–4], as well as to develop and test scenarios for implementing land-use and land-cover changes [5]. In most of these studies, the relationship between human population dynamics and past climate fluctuations is evident. Nevertheless, conceptual and methodological approaches from the theory of population dynamics (TPD) could be used to gain further insights into this phenomenon. Moreover, the application of formal methods of modelling provides a context where hypotheses can be tested against empirical evidence.

*Proc. R. Soc. B* **287**: 20200662

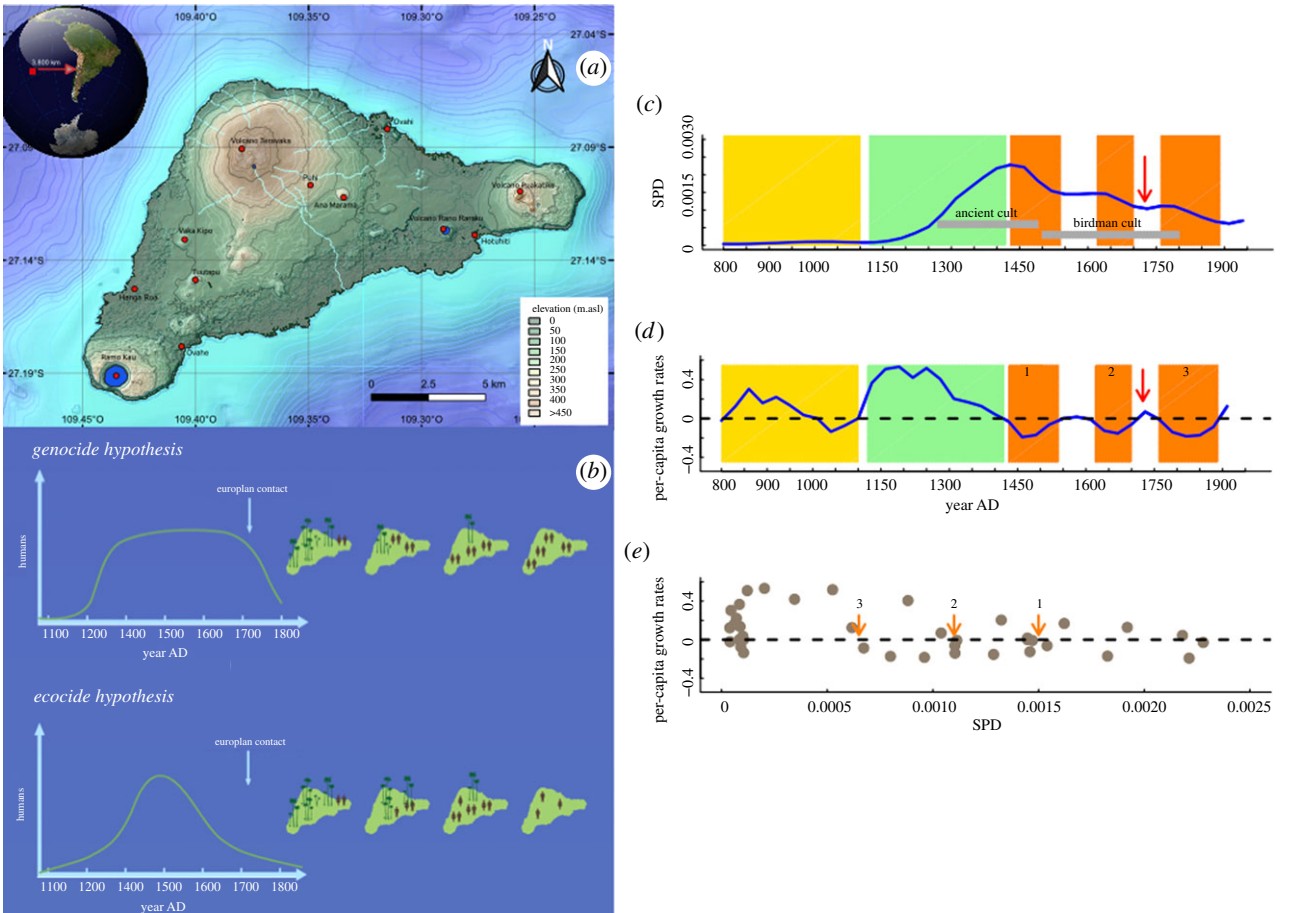

**Figure 1.** Population data and hypotheses description. (*a*) Map of Rapa Nui showing the major topographic features of the island. (*b*) Two hypotheses previously proposed for explaining the socio-ecological trajectory of Rapa Nui. *The ecocide hypothesis*: human population increased exponentially since its initial colonization. Human populations overshot the carrying capacity to support sustainable crops, and in consequence, the island was severely deforested. Such exhaustion of natural resources led to socio-demographic collapse. *The genocide hypothesis*: human population grew exponentially since approximately 1150 CE, reached the carrying capacity, but maintained a stable population size and sustainable crops despite deforestation. The collapse occurred after European colonization [14]. (*c*) Reconstructed SPD for Rapa Nui. Yellow and green vertical bars denote periods of low and exponential population growths, respectively. Orange areas indicate periods with negative population growth rates. The red arrow indicates the arrival of Europeans. (*d*) Time-series for per-capita growth rates (blue line) showing the chronology and magnitude of the three demographic collapses (orange bars). (*e*) R function plotted against the reconstructed SPD (one generation before = 30 years). Orange arrows indicate equilibrium population sizes during the three declines. Departure >0 implies a stronger relationship between SPD values and per-capita growth rates. (Online version in colour.)

Since food producer population equilibrium levels are set by crop productivity and land area, explanations of climatic effects on these societies must rely on formal examinations of indirect effects that climate changes could exert on the food production. This is because climate affects the availability of limiting resources (food production), and in turn, it impacts the per-capita resource availability shared between individuals of a given population [6]. These non-additive effects are expected when the ratio (i.e. population/food production) defines the per-capita share of the resources and the competition strength. Under such scenarios, small changes in a climate factor could have large changes in population growth rates due to the coupled interaction between climate, food production and population size [6]. Previous studies have developed models of human and renewable resource dynamics in order to understand the potential role of overpopulation and resource depletion in agrarian societies [7–11], with particular emphasis in the Rapa Nui case [12]. However, to the best of our knowledge, the role of climate non-additive perturbations on the population dynamic of past food producer societies has rarely been explored [13].

Here, we apply simple theoretical population dynamic models for deciphering the interaction between climatic variability, crop productivity and the long-term dynamics of the Rapa Nui population from the easternmost fringes of Polynesia (figure 1*a*). The Rapa Nui example represents one of the most controversial cases for the decline of an ancient farming society [15], which has been predominantly explained by invoking an anthropogenically driven ecological catastrophe. This 'ecocide hypothesis' assumes a punctuated demographic crash caused by the abrupt reduction and replacement of native palm forest by grasslands, ultimately driven by unchecked population growth coupled with social instability as the per-capita food productivity decreased (figure 1*b*) [16–18]. Alternatively, a 'genocide hypothesis' assumes that the Rapa Nui society did not collapse until European contact in the early eighteenth century (figure 1*b*) [19]. Past climate fluctuations have also been invoked [20], but to date there is no evidence showing any direct link between climate change and societal collapse on Rapa Nui.

Recent studies [21–23] challenge the predominant causal explanations for large-scale demographic downfall and

emphasize the potential roles of climate, anthropogenic pressures and the climate–human relationship. In fact, three main societal crises appear to have occurred in Rapa Nui. The first is dated to 1450–1550 CE (coeval to the Little Ice Age—LIA) and has been attributed to climate–cultural feedbacks and synergies. The second crisis is documented by the nineteenth century due to the introduction of epidemic diseases and the slave trade. Another, less apparent population decline occurred between the arrival of the first Europeans in 1722 CE and in 1774 CE for as yet unknown reasons. By combining independent sources of palaeoclimate, palaeoecological and palaeodemographic data, we argue that the use of the TPD framework represents an important advance in understanding the population dynamics of not only Rapa Nui people, but also other prehistoric food producer societies. We have developed and tested explicit population dynamic models (PDMs) that integrate hydroclimatic, demographic and ecological factors. By this means, we provide an alternative interpretation for the local population dynamic as well as new insights into the mechanisms underpinning the long-term trajectory of the human–environment interaction in the island.

# 2. Radiocarbon dates as data of prehistoric human population dynamics

Archaeologists are using large samples of [14]C dates to estimate human demographic trends in prehistoric populations [2–4]. The use of these data for making inferences about population dynamic patterns is not without methodological challenges [5,24]. For example, [14]C dates may be biased due to effects of the number/distribution of archaeological sites, sample size, calibration process, taphonomy and the type of economic or ritual activity [24,25]. Radiocarbon dates might be an estimate of the quantity of energy consumed by a prehistoric human population [24]. Although both population size and economic complexity have effects on energy consumption, population size explains more of the variability in energy consumption than economic complexity [24]. In fact, Freeman *et al.* [24] state that 'The sub-linear scaling of population size with energy consumption suggests that current approaches to interpreting date frequencies systematically misestimate population size over a given interval of time and growth rates.' Because we are interested in deciphering the relative, not the absolute, changes of Rapa Nui population growth rates and population size in a relatively short period, the use of SPD time series is justified, instead of other statistical approaches such 'tempo plots' [26] that are aimed to evaluate and constrain the extent (rhythms) of specific archaeological events (e.g. monumental construction in Rapa Nui [15]).

We have omitted obsidian hydration dates (OHDs) from our analysis. This decision lies with our need for data validated as proxies for population levels. More importantly, we aim to develop a framework based on a common palaeodemographic proxy to compare fairly dynamics among prehistoric populations from different cultural contexts or chronological phases. Unlike [14]C dates, the availability of OHDs is case-specific as it depends on the preservation/recovery of certain artefacts (lithic artefacts) from the archaeological record and is subject to spatio-temporal variations in technologies, subsistence strategies or cultural behaviours.

Although the absolute number of radiocarbon dates from Rapa Nui would be relatively small (244 [14]C dates), the number of dates per unit area per unit time is greater than or equal to previous studies [3]. For example, at the local scale, a study with a very high number of [14]C dates on different regions of Britain and Ireland during the mid- and late Holocene showed a range of sampling intensity between 0.13 and 0.26 dates × 100 yr$^{-1}$ × 100 km$^{-2}$ [3], whereas the sampling intensity at Rapa Nui is 0.19 dates × 100 yr$^{-1}$ × 100 km$^{-2}$. In terms of the ratio number of archaeological sites/geographic area, this study from the British Island and Ireland has a spatial sampling intensity of 1.8 sites × 100 km$^{-2}$ [3], whereas our spatial sampling intensity is 58 sites × 100 km$^{-2}$. Therefore, as both cases are equivalent in both sampling intensity and spatial representativeness, the employment of the SPD methodological approach is fully justified.

# 3. Material and methods

## (a) Palaeodemographic data

The human population size proxy is based on the SPD obtained from 244 archaeological radiocarbon dates. This proxy assumes that temporal variations in the accumulation of archaeological [14]C dates on a given region reflect human energy consumption/production [24], which in turn is a function of regional demographic patterns [27]. To estimate how Rapa Nui populations fluctuated over time, our SPD is based on 244 radiocarbon dates obtained from 95 archaeological sites (electronic supplementary material, table S4) reported by Mulrooney [28] ($n = 218$) and Commendador *et al.* [29] ($n = 26$). We have accepted the quality and reliability control of the chronological determinations presented in [29]. The resulting palaeodemographic time-series encompasses Class 1 and 2 dates compiled by Mulrooney [28], omitting 32 entries that yield modern calibrated ages (less than 125 [14]C years BP). Although the timing for the initial human colonization of Easter Island is still controversial—with conservative estimates around 1200 CE [15,19]—the oldest cultural date considered here is $1180 \pm 230$ [14]C yr BP (on wood charcoal) from a residential context [28]. Our time-series thus reproduces palaeodemographic patterns since 890 CE, the earliest established chronology for the initial peopling based on palaeoenvironmental and archaeological evidence [21,22,28].

Terrestrial ($n = 214$) and marine ($n = 4$) dates were calibrated at the 2-sigma confidence interval using SHCAL13 and MARINE13 curves, respectively. A local reservoir correction of $-83 \pm 34$ years [30] was applied on marine [14]C-ages assuming the 100% contribution of marine carbon. Twenty-six radiometric determinations on dentine–collagen samples were calibrated using estimated proportions ($\delta^{13}$C interpolation end-members) of marine carbon in human diet [29] (electronic supplementary material, table S4).

In practice, the SPD of calibrated [14]C dates was obtained by implementing the framework proposed by Crema *et al.* [31] in the rcarbon package for R, which can statistically assess and overcome any methodological biases (oversampling, calibration effect and taphonomic loss). To account for the overrepresentation of well-dated sites, normalized calibrated dates from a same archaeological site were aggregated into bins of 50 years. Any biases due to calibration were tackled by applying a 100-year rolling mean to produce the normalized SPD presented in electronic supplementary material, figure S1. Finally, a linear null model was fitted via 1000 Monte Carlo simulations for detecting statistically significant palaeodemographic fluctuations from the obtained normalized and smoothed SPD curve. By this means, magenta (blue) vertical bars in electronic supplementary material, figure S1 represent significant positive (negative) deviations from the critical 95% confidence area for the null model, and in turn, genuine population rises (falls) that cannot be explained by taphonomic loss and/or the overall trend of increasing population growth observed for Rapa Nui over the

past 800 years (figure 1c). Before the implementation of PDMs, the normalized and smoothed SPD time-series was sectioned into time-step intervals of 30 years in order to only capture large population trends and avoid high-frequency noise source of variability. Then, our models were fitted with SPD data taken at these regular intervals since 890 CE onwards.

## (b) Palaeoclimate data

Precipitation at Rapa Nui depends on the westerlies winds crossing the island, which in turn are controlled by the interplay of the South Pacific Anticyclone (SPA), the South Pacific Convergence Zone (SPCZ) and the westerly storm tracks [32]. The interaction between these three large systems determines the climate on the island. Seasonally, the weakening and northward migration of SPA during fall and winter causes an increase in precipitation during the April–June period, allowing the storm fronts associated with the westerlies winds pass over the island. During the summers, SPA is displaced southwards blocking the westerly storm fronts north of 34° S [33] and areas such as the Rapa Nui Island receive minimum precipitation [32]. Additionally, a relevant component of annual precipitation has been attributed to land–sea breeze convection storms [34].

The role of ENSO in driving short-term Holocene climate variations in Rapa Nui has been hotly debated as model simulations, and raw statistical analyses on observational data suggest inconclusive or non-significant correlations at interannual and centennial scales [34,35]. Recent reanalysis data assimilation for the period 1960–2014 CE [36], however, indicates that ENSO could provoke local precipitation and temperature anomalies brought about by perturbations on the main atmospheric circulation structures that modulate interannual and seasonal variability over the Southern Pacific Ocean. ENSO modifies the position of SPA, SPCZ and Subtropical Jetstream (STJ). During El Niño (La Niña) events, the SPCZ experiences a northeast (southwest) displacement that leads to increased (decreased) convective activity and rainfall amounts over Rapa Nui [36]. This warm (cold) phase also results in increased (suppressed) frontal precipitations facilitated by the weakening (intensification) and northward (southward) migration of the Pacific High, as well as through the equatorward (poleward) incursion of mid-latitude westerlies. Here, we performed correlation analyses between the Southern Oscillation Index (SOI) and 54-year time-series of mean annual rainfall in Rapa Nui (1961–2015 CE). Our results point to a major influence of La Niña episodes on the local hydroclimate. A weak but significant negative correlation exists between rainfall and the SOI ($r = -0.34$, $p = 0.016$, d.f. = 49, $t = 2.50$), indicating that positive SOI values (La Niña) are associated with decreased rainfall. This implies that Rapa Nui hydroclimate is sensitive only to cold ENSO phases, probably due to its influence on the position of the SPA, the SPCZ and STJ, as previously suggested in [32,37,38].

These circulation patterns can operate at several time scales—from annual to millennial [36,38]—and imply that reconstructions for past changes in ENSO activity can serve as a proxy for local hydroclimate conditions in Rapa Nui. Our selected proxy for ENSO-driven changes in the local hydroclimate is the 2000-year reconstruction for the SOI index by Yan et al. [39] (electronic supplementary material, table S5).

## (c) Model conceptualization for hypotheses

The basic dynamic model (equation (3.1)) predicts that after the initial colonization around 1100–1280 CE [15,28] population grew exponentially until equilibrium population size, which is determined by land area and food productivity [37,40,41]. The starting point is the exponential version of the general logistic equation [6],

$$N_t = N_{t-1} \cdot e^{r_N \cdot [1-(N_{t-1}/K)]}, \tag{3.1}$$

where $N_t$ denotes the human population size at time $t$, $r_N$ is a positive constant representing the maximum per-capita reproductive rate and $K$ is a constant for competition intensity and resource depletion represented by the availability of crop land and productivity (i.e. equilibrium population size) [6].

We assume that palm forest cover is a proxy for the potential agriculture area available for food production and food productivity; this assumption is consistent with the general pattern of association between palm forest clearance and the increase in charcoal influx and soil erosion in Polynesian islands [42,43] and in the particular case of Rapa Nui [23,37,41,44]. This dynamic can be described by a generalized exponential discrete logistic model for the human ($N$) population [6,45],

$$N_t = N_{t-1} \cdot e^{r_N[1-(N_{t-1}/K(F_{t-1}))]}, \tag{3.2}$$

where $r_N$ is a positive constant for maximum per-capita reproductive rates and $K(F_{t-1})$ is a function of the palm forest cover used as a proxy of resource availability (e.g. land use, ecosystem services and crop yields) [6].

Meanwhile, under a human population dynamic driven by the interaction between climate and crop production, the growth rate depends on the combined effect of recurrent droughts and anthropogenic pressures imposed by large human populations. Under this hypothesis, the human population ($N$) is described with the following model:

$$N_t = N_{t-1} \cdot e^{r_N[1-(N_{t-1}/K(C_{t-1}))]}, \tag{3.3}$$

where $N_t$ and $r_N$ are the same parameters as in equation (3.2), and $K(C_{t-1})$ is function of climate variability [6]. A detailed description of model parameters is depicted in electronic supplementary material, table S1.

Our starting point is to use the $R$ function [45] as a central element for connecting models and quantitative palaeodemographic, vegetational and palaeoclimate data. The realized population per-capita rate of change for a given interval of time, $R$, is defined as the value over the same time interval estimated by subtracting the logarithmic count at the first sampling date from that at the second. For human populations, this is $R = \log_e N_t - \log_e N_{t-1}$, where $N_t$ is the human population size (SPD data) at time $t$. In this study, the time interval used for human population analyses was 30 years, which represents a reasonable generational time for food producer populations [46]. For analytical convenience, we write equations (3.2) and (3.3) in terms of the per-capita rate of 'population' change:

$$R_t = r_N \left[ 1 - \left( \frac{N_{t-1}}{K(F_{t-1}, C_{t-1})} \right) \right]. \tag{3.4}$$

The chronologically well-constrained record of pollen percentages of palm from Raraku lake sediments [21] provides a time-series for local vegetation dynamics since 800 CE (electronic supplementary material, table S5). Thus, we defined a standardized measure of vegetation cover over time obtained from the perceptual representativeness of pollen types between consecutive $^{14}$C dates. This response variable was used in simple regression linear models for fitting explanatory variables of local palm forest cover dynamics. Thus,

$$F = \alpha + \beta \cdot N + \gamma \cdot C + \omega \cdot (N \cdot C) + \varepsilon, \tag{3.5}$$

where $F$ denotes the palm tree pollen %, $N$ is the SPD mean values of the previous 30 years, $C$ is the reconstructed SOI index and the term $N^*C$ represents the interaction between human population and climatic conditions. The interaction term assumes that human deforestation rates increase/decrease during periods of harsh/favourable climatic conditions for crop productivity.

Equation (3.4) was fitted through nonlinear least squares using the nls (nonlinear least squares) library in the R platform, assuming (i) a constant carrying capacity, (ii) a linear effect of the interpolated palm pollen percentage, (iii) a linear effect of reconstructed SOI index and (iv) both effects combined (electronic supplementary material, table S2), and equation (3.5) was fitted using simple linear regression models. We compared the goodness of fit between models by measuring the Akaike information criterion corrected for low sample size, AICc. Models with lowest AICc values were selected. Simulations provided the mean for evaluating the capacity of models for describing the observed dynamic of the SPD trends. Simulations were fitted considering the first observed value of the time-series, and then, running the algorithm using each model with their estimated parameters to obtain the simulated values. Uncertainty on SPD estimates were incorporated by resampling a joint multivariate normal distribution of the estimated parameters considering the asymptotic distribution of maximum-likelihood estimates. The 95% of the confidence intervals (CIs) of SPD time-series were obtained using the percentile method based on 10 000 iterations. In addition, values of the coefficient of prediction between observed and predicted values of each model were calculated to evaluate the predictive performance of the models using the following equation: $R^2 = 1 - \sum (Y - X)^2 / \sum (X - \text{mean } (X))^2$, where $Y$ is the predicted value of the model and $X$ is the observed value [47].

PDMs have a long history in ecological theory and modelling and provide a simple general framework for analysing and interpreting human population growth data. The PDMs developed here (equations (3.1)–(3.5)) are used to evaluate the expected predictions from the alternative ecological scenarios. The genocide hypothesis will be supported by a basic model of population growth without feedback between people and the limiting island resources, a simple logistic model with a stable population size before the European contact (equation (3.1)). Under the 'ecocide' hypothesis, the human population dynamics would be function of forest palm cover as a proxy of the available land for agriculture, and the change in palm pollen percentage will be negatively affected by human population size. On the other hand, if long-term climatic variability affected crop productivity and land use, the proper model structure is one where climatic variability and palm forest cover are proxies of limiting resources for human dynamics, while both human population size and climate will influence the palm pollen percentage decrease during the studied period. These testable models were fed with the 660-year palaeodemographic time series previously described as well as with time series for local vegetation dynamics and ENSO-induced hydroclimate anomalies (electronic supplementary material, tables S5 and S6).

## 4. Results

The SPD values obtained for Rapa Nui suggest that human energy consumption was very low from 800 to 1100 CE (figure 1c), which is indicative of a stationary phase with reduced demographic pressures on the island or even no stable human colonization. This phase appears as a statistically significant negative deviation against the lineal null model (electronic supplementary material, figure S1). In contrast, [14]C-dated densities increased quickly at 1150 CE, and different peaks and troughs of values are recognizable in the SPD curve from 1300 CE onwards (figure 1d; electronic supplementary material, figure S1). The resulting per-capita growth rate time series shows three different phases in the SPD dynamic of the Rapa Nui population over the last 800 years. First, a slow growth rate during the period 800–1100

CE (phase 1, figure 1d), followed by accelerated population growth from 1100 to 1400 CE (phase 2, figure 1d). These two phases encompass the statistically significant period of high density in [14]C dates observed between 1250 CE and 1530 CE (phase 3, electronic supplementary material, figure S1). The third phase characterizes three decrease–increase cycles (figure 1d,e). Two of these cycles (1430–1550 CE and 1640–1700 CE) occurred before European contact (cycles 3a and 3b; figure 1d,e). Compared with the null model, the first pre-European decline represents a significant negative deviation associated with a high amplitude negative trend that characterizes such peak in the SPD by approximately 1420 CE (electronic supplementary material, figure S1). While the remaining two declines extend throughout the overall low-density phase, they occur against a backdrop of significant negative deviation since 1660 CE onwards (electronic supplementary material, figure S1).

Per-capita growth rates are inversely related to SPD trends (figure 1e), suggesting the action of negative feedback processes in the dynamics of past Rapa Nui populations. This relationship is unclear, however, for the period 800–1100 CE due to the noisy nature of the palaeodemographic data (i.e. low-chronological resolution), which is probably associated with an actual later colonization of the island [19]. We vetted these inconsistent SPD values, as well as data from after European contact (greater than 1760 CE) from our subsequent analyses (figure 1e).

The coefficient of prediction between the basic model trajectories and data explains 61% of the observed variation (electronic supplementary material, table S2). In contrast, a model that includes lateral perturbation effects on carrying capacity [6] using palm tree pollen (model 2, electronic supplementary material, table S2) or the SOI reconstructed variability impacts (model 3, electronic supplementary material, table S2) significantly improves the prediction coefficients to 90% as well as model plausibility (Akaike weights; electronic supplementary material, table S2). Palaeoecological data indicate that from 1100 to 1760 CE, there was a clear and accelerated decrease in palm tree cover on the island, and palaeoclimate records show a coeval increasing trend towards positive SOI values (i.e. a 'La Niña-like' conditions) (figure 2a,b). Model simulations fed explicitly by available palaeoenvironmental information thus provide strong empirical support for a logistic population growth with a variable carrying capacity driven by land-use and hydroclimate to explain observed patterns from 1130 to 1760 CE. In fact, when SPD data are expressed as ratios of either palm pollen percentage (figure 2c) or SOI (figure 2d), we find a common negative linear $R$ function, which provides additional evidence for the feedback relationship between land use, demography and climate in driving Rapa Nui population dynamics. Both models show that lateral perturbation effects either by palm tree cover, SOI or both can better predict population dynamics compared with models that do not account for such exogenous effects (figure 3a–d).

Furthermore, palm tree pollen % dynamics in Rapa Nui are best explained by the negative effect of the human population, supporting the idea that pressure for farming land was the main driver of the deforestation process [23,41,44,48] (electronic supplementary material, table S3). Stone mulching farming and other agriculture features to assure food consumption prevented further soil and plant erosion because of wind and rain action [49–52].

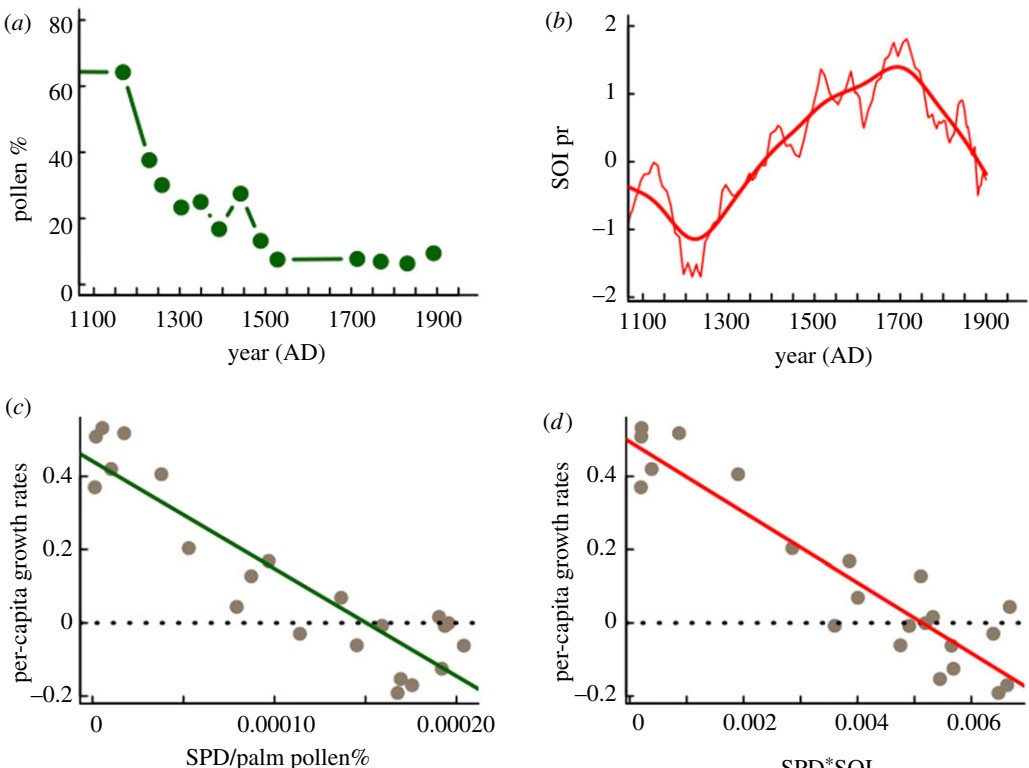

**Figure 2.** Palaeoenvironmental data. (*a*) Time series for the percentage of palm pollen (green line and dots) from the Raraku lacustrine record covering the period 1100–1960 CE [21]. (*b*) SOI reconstruction (SOI$_{pr}$), annually resolved (red thin line) and smoothed time series [39]. (*c*) Relationship between per-capita growth rates and the SPD per unit of palm pollen % for the same period (line fitted by linear regression; $y = 0.44 - 2935x$, $F_{1,20} = 97.23$, $p < 0.0001$, $r^2 = 0.82$). (*d*) Relationship between per-capita growth rates and the SPD by the unit of the Southern Index Oscillation for the same period (line fitted by linear regression; $y = 0.49 - 96.25x$, $F_{1,20} = 126.3$, $p < 0.0001$, $r^2 = 0.86$). (Online version in colour.)

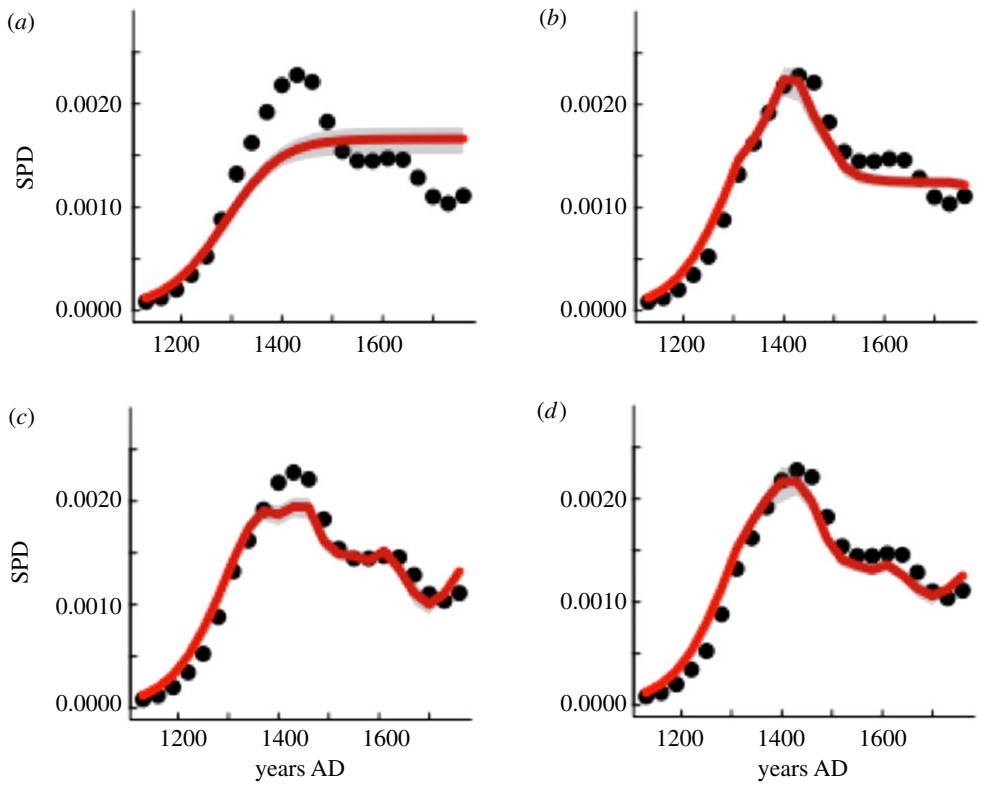

**Figure 3.** Comparisons between model-predicted trajectories (lines) and observed SPD data (points) at Rapa Nui. Estimated (black dots) and predicted (red line) SPD values for the period 1130–1760 CE. Grey shaded area shows 95% confidence intervals of model predictions based on Ricker logistic population models: (*a*) model 1, (*b*) model 2 (palm cover effects), (*c*) model 3 (climatic effects; SOI) and (*d*) model 4 (SOI and palm cover effects). See electronic supplementary material, tables S1 and S2 for details. (*d*) These models support the hypothesis that the Rapa Nui experienced major demographic changes well before the arrival of European colonizers. More importantly, this trajectory was represented by a very simple population model where carrying capacity is a function of either palm tree cover and climate, and offers an alternative to the simple dichotomy of 'ecological self-destruction' versus the 'idyllic equilibrium' with nature. (Online version in colour.)

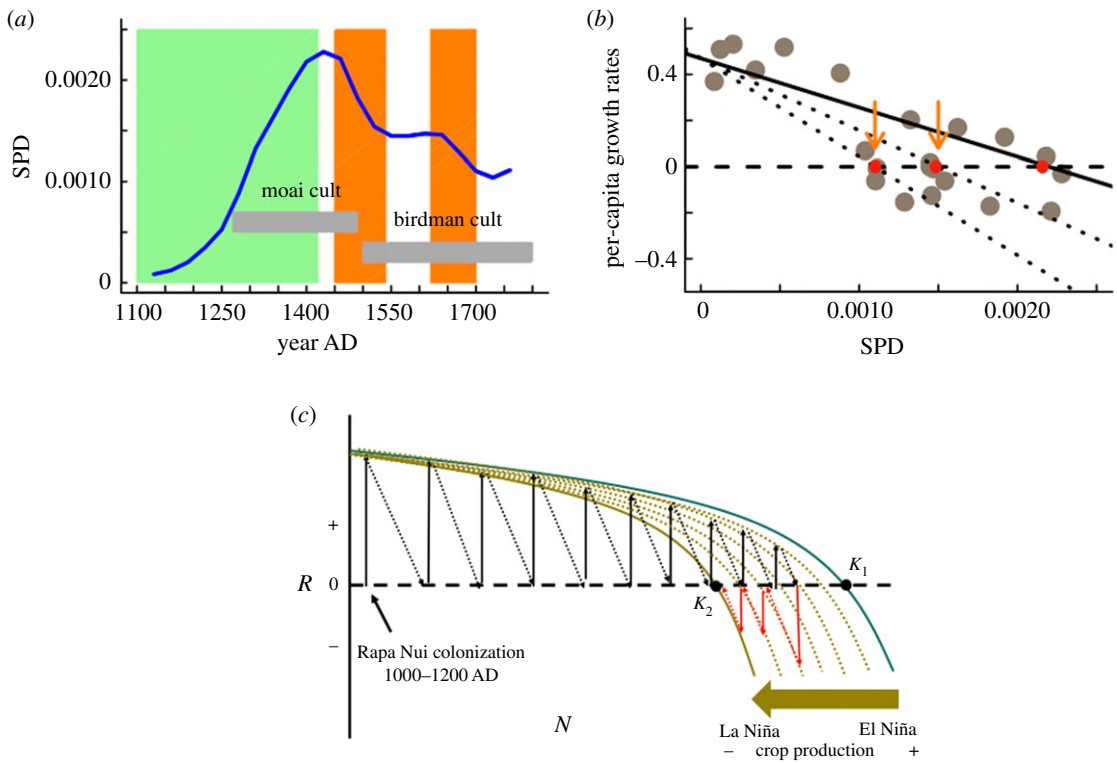

**Figure 4.** Graphical representations for the best model. (*a*) The observed population dynamic pattern from SPD data. (*b*) The effect of lateral perturbations caused by decreased crop production (or available land) either from palm tree cover or ENSO-induced hydroclimate anomalies. *K* represents the first equilibrium population size, and orange arrows denote reconstructed consecutive population declines. Solid and dotted lines are the graphical representations of the lateral perturbation effect. (*c*) Gradual changes in the island carrying capacity caused by a lateral perturbation process. The population trajectory of the Rapa Nui population in terms of the relationship between per-capita growth rates (*R*) and population size (*N*). The green solid curve represents the population logistic model with the carrying capacity $K_1$ at the initial colonization (1100–1200 CE). The gradual deterioration of the carrying capacity (dotted brown lines) is driven by negative hydroclimatic changes that lead to low crop productivity, intense land use and deforestation and in turn a lower equilibrium population size, $K_2$. Solid and dashed arrows illustrate demographic trajectories since the colonization (black arrows positive growth rates) to the collapses (red arrows negative growth rates). (Online version in colour.)

## 5. Discussion

Population analysis of the prehistoric Rapa Nui time series suggests that long-term climatic variability (e.g. SOI) and palm tree cover are proxies of the island's carrying capacity. In fact, a simple model appears to describe the dynamics of the human population in Rapa Nui quite well and is able to explain the increasing trend as well as population decline episodes that impacted during several generations, which we think can be defined as demographic collapses (*sensu* [53]).

Our results contradict recent work by DiNapoli *et al.* [15], which argues against pre-contact collapses in the island. The tempo-activity curve for the construction of megalithic structures reconstructed by DiNapoli *et al.* (fig. 5 in [15]) based on independent datasets is essentially—and paradoxically—equivalent to our normalized and smoothed SPD (figure 4*a*; electronic supplementary material, figure S1). These authors verify an exponential increase in the rate of construction since about 1300 CE, peaking at 1450 CE, and then this rate experiences a sustained slowdown since 1550 CE. We think that DiNapoli *et al.* [15] precluded collapses in the island by considering evidence for a totally different phenomenon: the 'continuities/discontinuities' in a given cultural tradition (i.e. the 'ahu moai' tradition), not a demographic process. Our analyses support a demographic expansion and collapse, as predicted by the 'ecocide' hypothesis [16–18] and consumer-resource models [12]. We think that the key driver in understanding the demographic and landscape changes in

Rapa Nui was the long-term forcing effect of climate on the island's per-capita resource base. In summary, climatic change interacts with the effects of human population size on land use.

Conversely, observed population boom and bust dynamics in Rapa Nui can be interpreted as the consequence of non-additive long-term effects of climate variability and palm tree cover on the limiting resources (land and food productivity) on human population dynamics, since equilibrium population size is set by which resource is in short supply [45] (figure 4*a*–*d*). This mechanism has been used for explaining the observed population collapses in Europe as a direct link between long-term climate change and carrying capacity [13]. In our results, demographic collapses of the Rapa Nui population resulted from the long-term effects of exogenous factors on the island's carrying capacity and in turn on the 'per-capita food supply'. This implies that the palm tree cover and climate may represent different proxies of the same limiting factors that control crop production in Rapa Nui. Human population size and energy consumption at the Rapa Nui society would be limited by food production, which depends on two processes: crop productivity and land availability. Our analyses are consistent with the hypothesis that climate variability influences crop productivity, whereas palm tree cover is a proxy of land use linked to the total cultivated area. Thus, the combination of positive population growth rates and a gradual increasing long-term trend in La Niña conditions (which causes droughts on the island) [37,54] could explain the extensive

deforestation as agriculture production increased the demand for farming lands, linked also to soil erosion and exhaustion. Intensified agricultural activities accelerated palm forest degradation further after 1200 CE [21]. Establishing causal links between climate and demographic change is always complex, and the effects of past hydroclimate variability on the island ecosystems have been widely debated [34,55]. Severe droughts during the second half of the Little Ice Age have been identified as the main driver of the demographic and cultural changes [56]. Indeed, the population crashes shown here were coeval with a positive trend of the SOI index during the period 1250–1700 CE on inter-decadal time scales, which have been recorded across the ENSO-sensitive regions in the Eastern Pacific [54]. Independently of the intensity/magnitude of any given environmental change (e.g. droughts), PDMs predict that the impact of exogenous variables on limiting factors is modulated by population size (non-additive effects), particularly through a per-capita resource share [6]. This means that even small perturbations in climate can trigger disproportionately large demographic responses when population sizes are near equilibrium, but much less so when such populations are in their exponential growth phase (figure 4c). Our results imply that climate effects should not be evaluated independently of population size when attempting to explain demographic changes.

The notion that forest degradation may have had a negligible impact on the crop production in the inherently nutrient-poor soils of Rapa Nui [14] has been challenged as it ignores concomitant soil erosion and loss of key ecosystem services (i.e. freshwater and nutrient cycling) [17,21,37,41]. Existing evidence suggests that forest clearance created widespread soil erosion between 1200 CE and 1650 CE [37,41]. More importantly, land use varied considerably after 1240 CE, driven by environmental constraints and variation in agricultural production [23]. The introduction of stone gardens after 1300 CE for agricultural production additionally helped to stop soil erosion [49–51]. These technologies lasted for less than 600 years until population density lowered to its minimum [50]. Precisely, SPD curves derived from OHD data reproduce consistent variations in land use for food production, with an overall expansion in intensity starting at 1240 CE and spatially heterogeneous slowdowns at approximately 1380–1430 CE and then since approximately 1660 CE [23].

Our results suggest that deforestation of the original palm forest of Rapa Nui is a consequence of the population pressure for stone farming land (rock gardens) [49,52] and not a cause in itself of population collapse. In fact, deforestation after the arrival of humans is a common phenomenon in the colonization of Polynesia [57] and elsewhere [58–61]. Moreover, the patterns of an increase in charcoal influx values for the period 1400–1700 CE, probably associated with an intensification of fire disturbance by humans [21], combined with the evidence of changes in the intensity of land use previous to the European contact [23], are consistent with the increase/decrease SPD dynamic.

The population decline from 1430 to 1550 CE (involving approximately four generations) could be in the origin of the posterior major socio-cultural changes in the island, such as from the moai cult to the birdman cult [62] (figure 4a), initiated around 1600 CE [63]. These changes show important investments in civil works for agriculture and water management, all of which represent profound changes in social organization and religion [48,64]. In fact, demographic and social collapses seem to be the rule across different farming cultures in the Pacific Islands [65]. Intense pressures on the island ecosystem services were required by the high-energy labour invested in the building and maintenance of the old socio-cultural system, materialized in the monumental architectural tradition of ancient ahu moai cult. Around 1500 CE, this ancient system shifted to a much simpler hierarchical political system and cult which required less energy and resource investment for its building and maintenance [63].

Although we do not test the relative role of high-energy perturbation events (e.g. volcano eruptions and tsunamis) in our analyses, these might have exerted positive synergistic impacts as documented for other Polynesian societies. For instance, the Kuwae eruption (1453 CE) matches with the major population crash detected in Rapa Nui (figure 4a), and it also is coeval to the interruption of the traditional Vanuatu–Tonga trade [66]. Palaeoenvironmental and documentary records suggest that deforestation and agricultural activities ceased by this period [21,67]. Whether the Kawae eruption played a role in driving a population collapse remains to be tested.

Understanding the relationship between human population dynamics and climate change is one of the major challenges for addressing the present environmental crisis and the possibility of a sustainable future [68]. Here, we demonstrate how the classical TPD is a useful framework for testing predictions of explanations for population changes in the Rapa Nui society. Finally, our results support the hypothesis that Rapa Nui experienced a long-term climatic trend towards drier conditions since the initial colonization of the island. This, concomitant with a gradual decrease in rainfall and island productivity, explains the process of expansion and demographic collapse experienced by the Rapa Nui society (figure 4c).

Population dynamic theory can thus be successfully applied to understand and explain population changes in prehistoric food producer societies. In particular, we highlight the use of the logistic model as it can take apart the simple dichotomy between the extremes of 'ecological self-destruction' versus 'idyllic equilibrium' with nature. Conversely, by undertaking profound adaptive socio-cultural practices, the Rapa Nui peoples were resilient for more than 1000 years on a tiny volcanic island in the middle of the Pacific Ocean, despite resource scarcity, overpopulation and climate change. Overpopulation and ongoing climate change are serious problems facing modern societies, and the historical trajectory described here for the Rapa Nui provides important insights regarding food security and ecological resilience for such populations.

Data accessibility. The data is available from the Dryad Digital Repository: https://doi.org/10.5061/dryad.bk3j9kd7j [69].

Authors' contributions. Conceptualization: M.L., E.M.G., S.A.E., C.L. and N.C.S. Data curation: E.M.G., N.C.-B., O.M., S.G., A.S. and S.P.-R. Formal analysis: M.L., S.A.E. and E.M.G. Investigation: M.L., E.M.G., S.A.E., C.L., C.M.S., N.C.-B., O.M., S.G., A.S., S.P.-R. and N.C.S. Methodology: M.L., S.A.E., N.C.-B. and E.M.G. Validation: M.L., E.M.G. and S.A.E. Visualization: M.L. and S.A.E. Writing—original draft: M.L., E.M.G., S.A.E., C.L., C.M.S. and N.C.S. Writing—review and editing: M.L., E.M.G., S.A.E., C.L., S.G., A.S., O.M. and S.P.-R.

Competing interests. We declare we have no competing interests.

Funding. M.L., E.M.G. and S.A.E. acknowledge financial support from the Center of Applied Ecology and Sustainability (CAPES;

ANID PIA/BASAL FB0002) and FONDECYT proposal no. 1180121—2018. We also thank ANID FONDAP and PIA grant nos 15110009 (to CR2) and AFB170008 (to the IEB) for additional funding.

Acknowledgements. This study was undertaken by the PEOPLE 3 K working group of the Past Global Changes (PAGES) project, which in turn received support from the Swiss Academy of Sciences and the Chinese Academy of Sciences.

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
