## [Reviewer comments · Proceedings of the Royal Society B: Biological Sciences]

Review History

RSPB-2020-0662.R0 (Original submission)

Review form: Reviewer 1

Recommendation

Accept with minor revision (please list in comments)

Scientific importance: Is the manuscript an original and important contribution to its field?

Good

General interest: Is the paper of sufficient general interest?

Good

Quality of the paper: Is the overall quality of the paper suitable?

Good

Is the length of the paper justified?

Yes

Should the paper be seen by a specialist statistical reviewer?

No

Do you have any concerns about statistical analyses in this paper? If so, please specify them explicitly in your report.

No

It is a condition of publication that authors make their supporting data, code and materials available - either as supplementary material or hosted in an external repository. Please rate, if applicable, the supporting data on the following criteria.

Is it accessible?

Yes

Is it clear?

No

Is it adequate?

No

Do you have any ethical concerns with this paper?

No

Comments to the Author

Please see the attached file (See Appendix A).

Review form: Reviewer 2

Recommendation

Accept with minor revision (please list in comments)

Scientific importance: Is the manuscript an original and important contribution to its field?

Good

General interest: Is the paper of sufficient general interest?

Excellent

Quality of the paper: Is the overall quality of the paper suitable?

Good

Is the length of the paper justified?

Yes

Should the paper be seen by a specialist statistical reviewer?

Yes

Do you have any concerns about statistical analyses in this paper? If so, please specify them explicitly in your report.

Yes

It is a condition of publication that authors make their supporting data, code and materials available - either as supplementary material or hosted in an external repository. Please rate, if applicable, the supporting data on the following criteria.

Is it accessible?

Yes

Is it clear?

Yes

Is it adequate?

Yes

Do you have any ethical concerns with this paper?

No

Comments to the Author

This is an interesting and provocative study which re-centres climate in a consideration of significant changes in the scale and continuity and dis-continuity of Rapanui's prehistoric monument-building societies and their subsequent later formulation. The paper will have a wide interest, given that Rapa Nui is recurrently used as a proxy for considering the ecological and cultural remits of societal 'collapse', particularly of monument-building societies. The paper is well written and rigorous in documentation of its data.

From a socio-cultural perspective, there are some concepts and terms used that incorporated perspectival assumptions that might be challenged; a brief exploration or awareness of this would be helpful to include. 'Collapse' is a loaded term potentially based on western perceptions of complex societies and what might be considered to be 'advanced'. Certainly, prehistoric Rapanui society has a major reconfiguration with colossal-stone monument construction eventually ceasing but with many other intricate, intense acts of construction continuing (rock gardens, petroglyphs, water management [as mentioned], pyramidisation of ceremonial monuments). Likewise 'decline' can be loaded word (particularly if attached or 'implied attached' to society or 'culture'), rather than say 'rapid change'. 'Agrarian' also requires comment - since non-crop resources (e.g. fish and chicken) are also part of the potential for securing the feeding of, and resilience of, Rapa Nui populations through periods associated with resource change. Also, how did agricultural adjustments (e.g. rock mulching and associated maintenance of the soil's minerals) impact on 'carrying capacity', given that carrying capacity relates to technology as well as environmental resources?

Comment might be provided on why only radiocarbon dates were used. How might they compare with the suggested patterns of population/human presence of Rapa Nui generated by obsidian hydration dating? The addition of a line or two on this might be helpful. The locales of the tabulated radiocarbon dates are given using term and place designations that are not available to a wider audience for assessment of their settlement, ceremonial or industrial context e.g. 'ahu', 'Maunga ...', 'East Poike', albeit that this will obviously be clear to those familiar with Rapa Nui.

Decision letter (RSPB-2020-0662.R0)

29-Apr-2020

Dear Dr Lima:

Your manuscript has now been peer reviewed and the reviews have been assessed by an Associate Editor. The reviewers' comments (not including confidential comments to the Editor) and the comments from the Associate Editor are included at the end of this email for your reference. As you will see, the reviewers and the Editors have raised some concerns with your manuscript and we would like to invite you to revise your manuscript to address them.

Research ethics:

Use of animals and field studies:

Please submit a copy of your revised paper within three weeks. If we do not hear from you within this time your manuscript will be rejected. If you are unable to meet this deadline please let us know as soon as possible, as we may be able to grant a short extension.

Best wishes,
Dr Locke Rowe
mailto:proceedingsb@royalsociety.org

Associate Editor
Board Member: 1
Comments to Author:

Thank you for your submission to Proceedings of the Royal Society. Two reviewers have positive things to say about your paper, as well as offering constructive criticism. One reviewer suggests that some of the language in the study and paper is loaded. The other reviewer would like more clarification of the model and its analyses so that it is fully reproducible by others. Please address the reviewers in a revision and response to reviewers. It is possible that the response will be considered in addition review, as well as by the associate editor.

Reviewer(s)' Comments to Author:

Referee: 1

Comments to the Author(s)
Please see the attached file.

Referee: 2

Comments to the Author(s)
This is an interesting and provocative study which re-centres climate in a consideration of significant changes in the scale and continuity and dis-continuity of Rapanui's prehistoric monument-building societies and their subsequent later formulation. The paper will have a wide interest, given that Rapa Nui is recurrently used as a proxy for considering the ecological and cultural relicts of societal 'collapse', particularly of monument-building societies. The paper is well written and rigorous in documentation of its data.

From a socio-cultural perspective, there are some concepts and terms used that incorporated perspectival assumptions that might be challenged; a brief exploration or awareness of this would be helpful to include. 'Collapse' is a loaded term potentially based on western perceptions of complex societies and what might be considered to be 'advanced'. Certainly, prehistoric Rapanui society has a major reconfiguration with colossal-stone monument construction eventually ceasing but with many other intricate, intense acts of construction continuing (rock gardens, petroglyphs, water management [as mentioned], pyramidisation of ceremonial monuments). Likewise 'decline' can be loaded word (particularly if attached or 'implied attached' to society or 'culture'), rather than say 'rapid change'. 'Agrarian' also requires comment - since non-crop resources (e.g. fish and chicken) are also part of the potential for securing the feeding of, and resilience of, Rapa Nui populations through periods associated with resource change. Also, how did agricultural adjustments (e.g. rock mulching and associated maintenance of the soil's minerals) impact on 'carrying capacity', given that carrying capacity relates to technology as well as environmental resources?

Comment might be provided on why only radiocarbon dates were used. How might they compare with the suggested patterns of population/human presence of Rapa Nui generated by obsidian hydration dating? The addition of a line or two on this might be helpful. The locales of the tabulated radiocarbon dates are given using term and place designations that are not available to a wider audience for assessment of their settlement, ceremonial or industrial context e.g. 'ahu', 'Maunga ...', 'East Poike', albeit that this will obviously be clear to those familiar with Rapa Nui.

RSPB-2020-0662.R1 (Revision)

Review form: Reviewer 1

Recommendation

Accept with minor revision (please list in comments)

Scientific importance: Is the manuscript an original and important contribution to its field?

Excellent

General interest: Is the paper of sufficient general interest?

Excellent

Quality of the paper: Is the overall quality of the paper suitable?

Good

Is the length of the paper justified?

Yes

Should the paper be seen by a specialist statistical reviewer?

No

Do you have any concerns about statistical analyses in this paper? If so, please specify them explicitly in your report.

No

It is a condition of publication that authors make their supporting data, code and materials available - either as supplementary material or hosted in an external repository. Please rate, if applicable, the supporting data on the following criteria.

Is it accessible?

Yes

Is it clear?

Yes

Is it adequate?

Yes

Do you have any ethical concerns with this paper?

No

Comments to the Author

This paper is much improved, and I recommend acceptance pending important clarifications below, especially with regard to the main conclusion. This paper will significantly contribute to the long-term analysis of human-population-climate interactions, and the methodology may be used in any region where the requisite data exists.

1. "The 117 audited of the radiocarbon dataset might allow to the minimization of these methodological 118 challenges [5]."

I am left totally guessing at the intended meaning of this sentence. I read the paragraph without it, and everything made sense. I recommend deleting.

2. "We are aware that the database size of radiocarbon dates from Rapa Nui would be relatively small 145 (244 14 C dates) compared with other studies [3]. However, sample size of radiocarbon dates and 146 number of sites must be compared with the time span and the spatial area studied."

Forgive the expression, but this language is very passive-aggressive. I recommend changing to something like:

Although the absolute number of radiocarbon ages from Rapa Nui is small (n=244), the number of ages per unit area per unit time is greater than or equal to previous studies. For example.....

3. "We think that the key driver in understanding the demographic 398 and landscape changes in Rapa Nui was the long-term forcing effect of climate on the island's per 399 capita resource base."

I don't think that this conclusion reflects the analysis, as least as presented in the SI. I do not mean to nit pick here. This is an important paper, and, in my view, the link between the conclusions drawn and the analysis needs to be clear.

Based on Table S2, the best fitting model includes both human induced changes in K by cutting or not palm trees and the climate effects on vegetation cover. I get this. This would indicate that both factors are important, together. Climate is no more key that population driven changes in land cover by palm trees. Absent any population growth and palm tree decline, the climate perturbation would not seem to be key at all. In other words, climate interacts with the state of the food production system. This seems to be supported by Table S3 where the best model for

predicting land cover change is a simple linear multiple of population (SPD) size. Though the equation that includes climate and population is not a bad fit either.

In sum, it seems that climate is key only in its interaction with the effect of population size on land cover. Perhaps I am missing something here. I don't mean to downplay the climate variable, just to make sure that a wide audience can follow the link from the analysis to the conclusion.

Decision letter (RSPB-2020-0662.R1)

27-May-2020

Dear Dr Lima

I am pleased to inform you that your Review manuscript RSPB-2020-0662.R1 entitled "The ecology of the collapse of Rapa Nui society" has been accepted for publication in Proceedings B.

The referee(s) do not recommend any further changes. Therefore, please proof-read your manuscript carefully and upload your final files for publication. Because the schedule for publication is very tight, it is a condition of publication that you submit the revised version of your manuscript within 7 days. If you do not think you will be able to meet this date please let me know immediately.

To upload your manuscript, log into <http://mc.manuscriptcentral.com/prsb> and enter your Author Centre, where you will find your manuscript title listed under "Manuscripts with Decisions." Under "Actions," click on "Create a Revision." Your manuscript number has been appended to denote a revision.

You will be unable to make your revisions on the originally submitted version of the manuscript. Instead, upload a new version through your Author Centre.

- 1) A text file of the manuscript (doc, txt, rtf or tex), including the references, tables (including captions) and figure captions. Please remove any tracked changes from the text before submission. PDF files are not an accepted format for the "Main Document".
- 2) A separate electronic file of each figure (tiff, EPS or print-quality PDF preferred). The format should be produced directly from original creation package, or original software format. Please note that PowerPoint files are not accepted.
- 3) Electronic supplementary material: this should be contained in a separate file from the main text and the file name should contain the author's name and journal name, e.g. `authorname_procb_ESM_figures.pdf`

All supplementary materials accompanying an accepted article will be treated as in their final form. They will be published alongside the paper on the journal website and posted on the online figshare repository. Files on figshare will be made available approximately one week before the accompanying article so that the supplementary material can be attributed a unique DOI. Please see: <https://royalsociety.org/journals/authors/author-guidelines/>

4) Data-Sharing and data citation

It is a condition of publication that data supporting your paper are made available. Data should be made available either in the electronic supplementary material or through an appropriate

repository. Details of how to access data should be included in your paper. Please see <https://royalsociety.org/journals/ethics-policies/data-sharing-mining/> for more details.

<http://datadryad.org/submit?journalID=RSPB&manu=RSPB-2020-0662.R1> which will take you to your unique entry in the Dryad repository.

Once again, thank you for submitting your manuscript to Proceedings B and I look forward to receiving your final version. If you have any questions at all, please do not hesitate to get in touch.

Sincerely,
Dr Locke Rowe
Editor, Proceedings B
<mailto:proceedingsb@royalsociety.org>

Associate Editor Board Member: 1

Comments to Author:

Thank you for your revision. I'd like you to pay attention to just one more issue: the reviewer feels like the conclusions still do not match the results. Please give their criticism on this point one more look. I agree that the conclusions can be tempered, and I'd like to hear your response to that point.

Reviewer(s)' Comments to Author:

Referee: 1

Comments to the Author(s)

This paper is much improved, and I recommend acceptance pending important clarifications below, especially with regard to the main conclusion. This paper will significantly contribute to the long-term analysis of human-population-climate interactions, and the methodology may be used in any region where the requisite data exists.

1. "The 117 audited of the radiocarbon dataset might allow to the minimization of these methodological 118 challenges [5]."

I am left totally guessing at the intended meaning of this sentence. I read the paragraph without it, and everything made sense. I recommend deleting.

2. "We are aware that the database size of radiocarbon dates from Rapa Nui would be relatively small 145 (244 14 C dates) compared with other studies [3]. However, sample size of radiocarbon dates and 146 number of sites must be compared with the time span and the spatial area studied."

Forgive the expression, but this language is very passive-aggressive. I recommend changing to something like:

Although the absolute number of radiocarbon ages from Rapa Nui is small ($n=244$), the number of ages per unit area per unit time is greater than or equal to previous studies. For example....

3. "We think that the key driver in understanding the demographic changes and landscape changes in Rapa Nui was the long-term forcing effect of climate on the island's per capita resource base."

I don't think that this conclusion reflects the analysis, as least as presented in the SI. I do not mean to nit pick here. This is an important paper, and, in my view, the link between the conclusions drawn and the analysis needs to be clear.

Based on Table S2, the best fitting model includes both human induced changes in K by cutting or not palm trees and the climate effects on vegetation cover. I get this. This would indicate that both factors are important, together. Climate is no more key than population driven changes in land cover by palm trees. Absent any population growth and palm tree decline, the climate perturbation would not seem to be key at all. In other words, climate interacts with the state of the food production system. This seems to be supported by Table S3 where the best model for predicting land cover change is a simple linear multiple of population (SPD) size. Though the equation that includes climate and population is not a bad fit either.

In sum, it seems that climate is key only in its interaction with the effect of population size on land cover. Perhaps I am missing something here. I don't mean to downplay the climate variable, just to make sure that a wide audience can follow the link from the analysis to the conclusion.

Author's Response to Decision Letter for (RSPB-2020-0662.R1)

See Appendix B.

Decision letter (RSPB-2020-0662.R2)

29-May-2020

Dear Dr Lima

I am pleased to inform you that your manuscript entitled "The ecology of the collapse of Rapa Nui society" has been accepted for publication in Proceedings B.

Open Access

Paper charges

Sincerely,

Appendix A

Review of "The ecology of the collapse of Rapa Nui agrarian society"

Summary: This paper compares the fits of population dynamic models to a paleo population proxy on Rapa Nui. Specifically, they compare a simple logistic population model with logistic population models in which K (carrying capacity) changes over time due to endogenous harvest pressure and exogenous climate change. The authors use summed probability distributions generated through the analysis of a high density sample of radiocarbon dates (as well as paleoclimate and ecological proxies) to evaluate the alternative population models. Their analysis supports an pre European population collapse/decline to a lower equilibrium due to the combined effects of endogenous harvest pressure and climate on palm tree cover (main determinant of soil fertility and arable land on the island?).

Recommendation:

This is an interesting paper that is consistent with earlier theoretical models (under cited by archaeologists) on the collapse of Rapa Nui populations. The paper makes an important contribution by using population dynamic models to develop hypotheses that might explain paleo population estimates from SPDS. I think that this is the kind of theory-driven work that increases the usefulness of radiocarbon analysis.

I recommend that the paper is accepted for publication following suitable, relatively minor revisions.

Substantive Comments:

(1) The paper would improve most by clarifying how the models operationalize or are different from the ecocide/genocide debate, which seems overly simplistic. I also think the impact of the paper would improve if connected with other theoretical models on agrarian society population and resource harvest dynamics. This model and data analysis exercise seems to correspond with such previous papers quite well, which is interesting and important from the perspective that converging results from both theory and data can give us more confidence in our knowledge claims.

I have provide some specific references and identified sections below where the authors might revise to clarify the connection/differences between their models and the ecocide/genocide hypotheses.

(2) The authors assume a lot of math background in their theory section. I think for wide audience of this journal, a bit more intuitive description of the models would be helpful. Also, how is equation 5 used? Was it a poor fit? Please clarify the connection between equation 5 and equations 2 and 3. I think this will be a source of confusion for many readers.

Along the lines of increasing accessibility, a table that defines key model parameters and links these to proxies could be helpful.

(3) Please clarify the places where the logic behind assumptions is a bit thin. For example, my minor points 4 and 5 below.

(5) The authors should analyze the sensitivity of their results to calibration effects. Perhaps use a 60 year time step in addition to the 30 year, which should be slightly less sensitive to suck and smear effects of the calibration curve. This would serve as a kind of robustness check on the results even though double the standard human generation time. Other radiocarbon specialists may have other ideas and concerns.

(6) I recommend more description of the data displayed in the SI and how to replicate their analysis. The commented r code linked to specific analyses in the paper would also be extremely useful for replicating the paper.

Minor Comments:

1. “the role of climate non-additive perturbations on the population dynamic of past agrarian societies has rarely been explored
79 [7].”

Maybe the conjunction of formal models and empirical analysis is rare, but many models explore the non-additive effects of climate in the context of resource use in agrarian societies. I think the paper would improve if connected with these models more explicitly:

e.g.,

Brander, J. A., & Taylor, M. S. (1998). The simple economics of Easter Island: A Ricardo-Malthus model of renewable resource use. *American economic review*, 119-138.

Puleston, C., Tuljapurkar, S., & Winterhalder, B. (2014). The invisible cliff: abrupt imposition of Malthusian equilibrium in a natural-fertility, agrarian society. *PloS one*, 9(1).

2. “The
115 audition of the radiocarbon dataset might allow to the minimization of these methodological
116 challenges [5]”

This sentence is very hard to understand. Is audition the intended word?

3. “To fit the population dynamic models, we used the smoothed 100-year rolling mean SPD data and
180 a time-step interval 30 years in order to only capture large population trends and avoid high
181 frequency noise source of variability.”

I don't quite follow what you did here. I think other readers will have the same problem. The sentence makes it sound like you sampled the 100 year rolling mean at 30 year intervals. Please clarify.

4. “We assume that palm forest cover is a proxy for the potential agriculture area available for crop
235 production or crop productivity.”

Please state why this is a legitimate assumption. This will improve the paper by making the rationale for using this proxy clear so that one can judge the strengths and weaknesses of the proxy. Is more trees = higher soil fertility; fewer trees = less soil fertility in a slash and burn system?

5. Lines 293 to 302. I recommend deleting the header. The paper would improve here if the authors can spend a little more space connecting the equations to the qualitative hypotheses that they are meant to make operational.

How does equation 5 fit in? Basically, palm cover is affected by both endogenous harvest dynamics (which is some linear combination of N) and exogenous climate dynamics. Makes sense. Why do C and N interact? Is it because bad climate years lead to more intense clearing to make up shortfalls and good years less intense clearing? If so, there seems to be an implicit assumption about human behavior that would be worth making explicit here. Or at least citing some ethnographic evidence that supports such a dynamic—which I think does exist. Farmers do try to make-up for climate induced shortfalls, if they have the requisite storage infrastructure, by growing more...

6. Table S2 does not include the interaction term from equation 5, why? Without the interaction term, the conclusion that

“ Furthermore, palm tree cover

346 dynamics in Rapa Nui are best explained by both a Malthusian and a climatic component, which in
347 this case is the negative effect that arises from the combined effects of human population size and
348 ENSO-driven droughts (Table S2)”

is hard to follow. Do your results suggest that the effects of the Malthusian component (harvest pressure) and climate are, in fact, separable, which would not support equation 5?

7. After reading your results, it seems that your analysis provides strong support for the dynamics proposed by Brander and Taylor

Brander, J. A., & Taylor, M. S. (1998). The simple economics of Easter Island: A Ricardo-Malthus model of renewable resource use. *American economic review*, 119-138.

Appendix B

May 27, 2020

Dear Locke Rowe
Proceedings of the Royal Society of Biological Sciences

Resubmission of our revised manuscript for a review paper for PRSB

I refer to your feedback on the last submission which I received by mail on May 27, 2020. Below we answer the last comments of the reviewer 1.

Referee 1:

Query 1: 'The audited of the radiocarbon dataset might allow to the minimization of these methodological challenges [5]'. I am left totally guessing at the intended meaning of this sentence. I read the paragraph without it, and everything made sense. I recommend deleting.

Reply: We agreed and deleted the paragraph.

Query 2: We are aware that the database size of radiocarbon dates from Rapa Nui would be relatively small (244 ¹⁴C dates) compared with other studies [3]. However, sample size of radiocarbon dates and number of sites must be compared with the time span and the spatial area studied.". Forgive the expression, but this language is very passive-aggressive. I recommend changing to something like: Although the absolute number of radiocarbon ages from Rapi Nui is small (n=244), the number of ages per unit area per unit time is greater than or equal to previous studies. For example.....

Reply: Agree and fixed. Now we wrote "Although the absolute number of radiocarbon dates from Rapa Nui would be relatively small (244 ¹⁴C dates), the number of dates per unit area per unit time is greater than or equal to previous studies [3]. For example, at the local scale, a study with very high number of ¹⁴C dates on different regions of Britain and Ireland during the mid and Late Holocene showed a range of sampling intensity between 0.13 and 0.26 dates x 100 y⁻¹x100 km⁻² [3], whereas the sampling intensity at Rapa Nui is 0.19 dates x 100 y⁻¹x100 km⁻²."

Query 3: "We think that the key driver in understanding the demographic and landscape changes in Rapa Nui was the long-term forcing effect of climate on the island's per capita

resource base." I don't think that this conclusion reflects the analysis, as least as presented in the SI. I do not mean to nit pick here. This is an important paper, and, in my view, the link between the conclusions drawn and the analysis needs to be clear. Based on Table S2, the best fitting model includes both human induced changes in K by cutting or not palm trees and the climate effects on vegetation cover. I get this. This would indicate that both factors are important, together. Climate is no more key that population driven changes in land cover by palm trees. Absent any population growth and palm tree decline, the climate perturbation would not seem to be key at all. In other words, climate interacts with the state of the food production system. This seems to be supported by Table S3 where the best model for predicting land cover change is a simple linear multiple of population (SPD) size. Though the equation that includes climate and population is not a bad fit either. In sum, it seems that climate is key only in its interaction with the effect of population size on land cover. Perhaps I am missing something here. I don't mean to downplay the climate variable, just to make sure that a wide audience can follow the link from the analysis to the conclusion.

Replay: We think that the meaning of the term "per capita resource base" is what reviewer commented, climate is interacting with the per capita food production system of the island. But, we modified this paragraph in order to be more explicit about this interaction and that is not just climate change. We wrote "We think that the key driver in understanding the demographic and landscape changes in Rapa Nui was the long-term forcing effect of climate on the island's per capita resource base. In sum, climatic change interacts with the effects of human population size on land use."

On behalf of the authors,
Best regards,